# Etiology, Predisposing Factors, Clinical Features and Diagnostic Procedure of Otomycosis: A Literature Review

**DOI:** 10.3390/jof9060662

**Published:** 2023-06-13

**Authors:** Mila Bojanović, Marko Stalević, Valentina Arsić-Arsenijević, Aleksandra Ignjatović, Marina Ranđelović, Milan Golubović, Emilija Živković-Marinkov, Goran Koraćević, Bojana Stamenković, Suzana Otašević

**Affiliations:** 1Medical Faculty, University of Niš, 18000 Niš, Serbia; milabojanovic@yahoo.com (M.B.); drsalea@yahoo.com (A.I.); marina87nis@gmail.com (M.R.); emilijazm@gmail.com (E.Ž.-M.); gkoracevic@yahoo.com (G.K.); bojana.stamenkovic.70@gmail.com (B.S.); 2University Clinical Center Niš, 18000 Niš, Serbia; milanpfc@gmail.com; 3Medical Faculty, University of Priština in Kosovska Mitrovica, 38220 Kosovska Mitrovica, Serbia; stale1995@gmail.com; 4Medical Faculty, University of Belgrade, 11000 Belgrade, Serbia; mikomedlab@yahoo.com; 5Public Health Institute Niš, 18000 Niš, Serbia; 6Institute For Treatment and Rehabilitation “Niška Banja”, 18205 Niš, Serbia

**Keywords:** otomycosis, epidemiology, etiology, predisposing factors, clinical features, diagnostic procedure, treatment

## Abstract

Otomycosis (OM) is a superficial fungal infection of the external auditory canal (EAC) with a worldwide prevalence ranging from 9% to 30%. Commonly, otomycoses are caused by *Aspergillus (A.) niger* complex and *Candida* spp. Other causative agents are yeasts of the genera *Cryptococcus* spp., *Rhodotorula* spp., *Geotrichum candidum*, dermatophytes (*Trichophyton mentagrophytes*), and non-dermatophytes molds (*Fusarium* spp., *Penicillium* spp., *Mucorales* fungi). The widest range of different species causing OM are found in the territories of Iran, India, China, Egypt, Mexico, and Brazil. Fungal infection of the EAC varies from mild to severe forms. It can be acute, subacute, or chronic, and is often unilateral, while the bilateral form is more common in immunocompromised patients. From an epidemiological point of view, tropical and subtropical climates are the most significant risk factor for the development of otomycosis. Other predisposing conditions include clothing habits, EAC hygiene practices, long-term antibiotic therapy, diabetes, and immunodeficiency. Since it is often difficult to distinguish otomycosis from an infection of a different origin, laboratory-based evidence, including standard procedures (microscopy and cultivation), is essential for diagnosis. For the treatment of this superficial fungal infection, there are no official therapeutic guidelines and protocols. However, many antifungals for local application, such as polyene, imidazoles, and allylamines, can be applied, as well as systemic antimycotics (triazoles) in severe forms of infection.

## 1. Introduction

Otomycosis is a common superficial fungal infection that affects the external auditory canal (EAC). The infection may present as acute, subacute, or chronic, and is typically unilateral, with the bilateral form being more common in immunocompromised patients. The infection is present globally, with prevalence ranging from 9% to 30% in patients with the signs and symptoms of EAC infection [1,2,3]. The most common causative agents of otomycosis are molds of the genus *Aspergillus* and yeasts of the genus *Candida*, particularly *Aspergillus (A.) niger* complex and *Candida (C.) albicans* [4].

Recent molecular methods and analyses have revealed 28 different species of black molds within the *Aspergillus niger* complex, with *A. tubingensis* being the most common cause of otomycosis, followed by *A. niger*. Although rare, the species *A. welwitschiae*, *A. awamori* and *A. foetidus* have also been reported as the cause of this infection [5]. Accurate molecular identification of these black fungi is crucial for the development of rapid diagnostic tests for *Aspergillus* otomycosis, and for determining appropriate treatment for different species within this complex. Apart from the molds mentioned, species of *A. flavus* complex are also commonly isolated, while *A. terreus*, *A. fumigatus*, *A. versicolor*, and *A. luchuensis* are less frequent culprits [6]. In the last year, *A. sydowii* was identified as a potential human pathogen in immunocompromised patients, with two reported cases in which the fungi were molecularly identified from the EAC material obtained during tympanomastoidectomy [7].

In addition to *C. albicans*, *C. parapsilosis* has been established as a common cause of otomycosis, particularly in Europe, where its prevalence is high. Other *Candida* species such as *C. lustaniae*, *C. guilliermondii*, *C. famata*, *C. tropicalis*, *C. krusei*, and *C. glabrata* have also been isolated and identified as etiological agents of the EAC infection, although with relatively low incidence [8]. Moreover, *C. auris*, an emerging and highly virulent fungal pathogen, has been detected as a colonizer of EAC skin and a potential cause of the infection. This is of major concern due to its multi-drug resistance and potential for dissemination [9]. 

Sporadically, otomycosis caused by yeasts of the genera *Cryptococcus*, *Rhodotorulla* spp. and *Geotrichum candidum* has also been reported [10]. Furthermore, EAC skin infection caused by fungi of the dermatophyte group (*Trichophyton mentagrophytes*) could easily occur during dermatophytosis via autoinfection. 

The external ear canal provides an ideal environment for the growth and reproduction of molds, and there has been an increase in cases of otomycosis caused by different species of non-dermatophyte molds, not only in immunodeficient but also in immunocompetent patients. 

A large number of fungal genera are found in nature, and one proposed classification divides infections caused by these non-dermatophyte molds into two broad groups: those caused by *Aspergillus* species, and those caused by non-*Aspergillus* molds. Non-*Aspergillus* infections can be caused by *Mucorales* fungi, hyalohyphomycetes (bright pigmented fungi), and phaeohyphomycetes (dark pigmented or dematiaceous fungi). In the group of non-dermatophyte molds, fungi of the genera *Penicillium* [10], *Scopulariopsis*, *Chrysosporium* [11], *Fusarium* [12], *Alternaria*, [12], *Paecilomyces* [13], *Auerobasidium* spp., *Acremonium* spp. [14], and species of the *Mucorales* group [10] are reported to be the cause of this infection [1,15]. Additionally, very unusual cases of otomycosis caused by *Saksenaea vasiformis*, a species of the order *Mucorales* [16], and *Scedosporium apiospermum* in an immunocompetent patient [17] have recently been published.

This narrative review aims to highlight the significant characteristics of otomycosis by reviewing the literature on the etiological–mycological, clinical and epidemiological features, diagnostic procedures, and previous experiences related to the treatment of this disease. Given the high prevalence of otomycosis worldwide and the lack of official and recommended treatments or diagnostic procedures, accurate identification of the causative agents is of the utmost importance in designing appropriate treatment strategies.

## 2. Materials and Methods

The databases Medline, Scopus, and Web of Science were searched as the primary sources of articles on the topic of superficial fungal EAC infection, using key terms such as “otomycosis”, “mycotic otitis externa” “epidemiology”, “causative agents”, “fungal species”, “diagnostics”, and “treatment”. Additionally, Google Scholar was searched, and the references of identified articles were hand-searched. To eliminate duplicate articles and articles that were not related, the search results were initially checked based on the title and abstract by one of the reviewers (AI). A more detailed review of the title, abstract, and subsequently the full text of retrieved articles was carried out by two independent reviewers (AI and MS). Disagreements were resolved through discussion. The protocol for this review has not been previously registered or published.

One reviewer extracted the following data: continent, country, number of patients, gender, age, causative agents, diagnostic procedures, clinical examination findings, treatment, and risk factors for otomycosis. The data extracted from the selected studies were presented, tabulated, and compared narratively. Pooling the studies statistically was not possible because of the heterogeneity in study designs, patient characteristics, and outcomes. The purpose of a narrative review is to identify research gaps within a specific area and provide a comprehensive summary of the evidence related to a research problem. This provides a general and accurate guide to what is already known about this subject and creates a framework for further research. However, in the undertaking of a narrative review, we should be aware of selection bias. 

## 3. Pathogenesis of Otomycosis

The external auditory canal is has a hollow cylinder-like shape; it is approximately 2–2.5 cm in length, enclosed by the tympanic membrane on its proximal end. Its other end is unobstructed and directly exposed to the external environment [18]. The inner surface of the canal is lined with skin, which also covers the outer side of the eardrum and continues onto the surface of the auricle. Intact skin acts as a mechanical barrier, preventing the penetration of microorganisms from the external environment. It also contains modified apocrine sweat glands that secrete cerumen, which has hydrophobic properties and prevents water retention in the ear canal. Cerumen forms a protective layer on the skin’s surface and exhibits antimicrobial properties due to its low pH, creating unfavorable conditions for pathogen development [19,20].

The surface of healthy EAC skin harbors various microbial species, including St*aphylococcus* spp., *Corynebacterium* spp., *Bacillus* spp., *Streptococcus* spp., Gram-negative bacilli (e.g., *Pseudomonas aeruginosa*, *Escherichia coli*), as well as fungi, predominantly of the *Aspergillus* and *Candida* species [21]. The saprophytic nature of these microorganisms can become pathogenic if the balance between bacterial and fungal growth is disrupted, especially if non-specific and specific body defense mechanisms are compromised [22,23].

Factors contributing to the development of otomycosis are categorized into environmental and host-derived. The warm and humid climate of tropical and subtropical regions is the most significant external risk factor, so it is not surprising that highest prevalence of otomycosis is recorded in these areas [24,25]. Host-derived risk factors include the specific anatomical features of the EAC, excessive cerumen secretion, local trauma of the ear canal, use of hearing aids with an occlusive mold, and immunocompromised health status. Otomycosis can also occur secondary to previous bacterial infection of EAC treated with topical antibacterial drugs. Additionally, the disease may result from autoinoculation (autoinfection) of the canal in patients with untreated dermatomycosis [20,25].

## 4. Clinical Presentation

Inflammation of the external auditory canal is characterized by an inflammatory process that affects the skin and subcutaneous tissue of the canal, with the potential to extend to the eardrum and auricle in severe cases [26]. The clinical presentation of this inflammation can vary from mild to severe forms that can threaten the patient’s life. In practice, there is a difference between localized and diffuse inflammation, which can be acute or chronic. Etiologically, inflammation can be of non-infectious or infectious nature. Non-infectious inflammation can be attributed to either a local cause, such as contact or allergic dermatitis, or to a systemic disease, such as seborrheic dermatitis or psoriasis [22,27,28]. One consideration is that skin diseases such as allergic or seborrheic dermatitis, as well as psoriasis, can create a conducive environment for the development of secondary infections caused by bacteria or, less commonly, fungi. Susceptibility to secondary infection may also be attributed to the yet insufficiently clarified pathogenic potential of certain fungi in initiating the infection. Bacteria have been identified as the most common cause of infectious inflammation, with a prevalence of up to 90%. *Pseudomonas aeruginosa* and *Staphylococcus aureus* are the most commonly found species, with a prevalence of 22–62% and 11–34%, respectively [29]. Fungi can also cause the inflammation of EAC, known as otomycosis.

Otomycosis is typically a benign and superficial infection, often asymptomatic in the early stages and not considered life-threatening. However, in certain situations, the infection may become recurrent due to the ideal environment for fungal growth provided by the EAC. It is important to note that the infection can spread and potentially affect the middle ear in approximately 10% of cases [30,31]. In severe cases, the infection can also spread to the surrounding soft tissues, including the parotid gland, and rarely, it may extend to the mastoid bone, temporomandibular joint, and skull base, potentially affecting the VII (facial), IX (glossopharyngeal), X (vagus), XI (accessory), or XII (hypoglossal) cranial nerves [32]. However, it should be noted that the most serious cases of otomycosis, with perforation of the tympanic membrane, involvement of the middle ear, or the entire temporal bone, are predominantly associated with immunodeficiency conditions [33].

Bone invasion or tissue damage generally does not occur in *Aspergillus* infection. In patients with *Candida* infection, the early stage of the infection is characterized mainly by exudation, and in the later stages, granulomatous inflammation becomes predominant (29). Species of the *Mucorales* genus can potentially penetrate blood vessels, causing thrombi, tissue infarction and leukocyte infiltration [34].

Infection is typically associated with a range of symptoms, including itching, otorrhea, and the sensation of ear canal blockage. Pain, which can vary in severity, headache, and tinnitus are also common. EAC edema and redness, desquamation of the epithelium, and impaired hearing are often clinical findings present in patients with otomycosis [1,35,36].

## 5. Diagnostic Procedure

Otomycosis diagnosis is primarily based on a patient’s history and clinical presentation, as well as an otoscopic examination of the ear canal and eardrum. However, additional tests such as microbiological analysis or histological examinations may be necessary to confirm the diagnosis and determine the causative organism, especially in severe or chronic cases of otomycosis. Imaging studies may also be used in rare cases to assess the extent of the infection or rule out other potential causes of symptoms [32,37].

Otomycosis is often difficult to distinguish from infections of different origins, especially in the case of diffuse external otitis. The particular challenge in diagnosis arises when there are mixed infections, where bacterial species such as *S. aureus*, *Pseudomonas* spp., coagulase-negative *Staphylococci*, or *Klebsiella* spp. are present along with fungi in the sampled material [38].

Laboratory-based evidence for otomycosis diagnosis is obtained using conventional methods. Microscopic examination is an easy, low-cost, and fast method that is irreplaceable for detecting fungi in patient material. Wet mount (native or with chloralactophenol or KOH) is still a convenient technique for screening and direct microscopy examination, providing prompt detection of fungal blastoconidia–yeast and pseudohyphal–hyphal forms in patient material [39]. Still, these methods have limitations, such as lower sensitivity, an inability to distinguish the species of causative agents, and difficulty in differentiating contaminants from infectious agents.

Cultivation, isolation, and identification of the fungus from the sampled material remain the gold standard for accurate diagnosis [12]. However, it is important to emphasize the necessity of serial mycological examinations (up to 3) for the accurate interpretation of cultivation-based mycological findings, in order to differentiate fungal causative agents from fungal microbiota, or transitory fungal flora. These are superior but time-consuming procedures that enable the determination of genetic and biochemical characteristics, identification of etiological agents, and antimicrobial susceptibility testing. 

For microbiological examination, it is important to conduct both bacteriological and mycological analyses. Cultivation of infectious agents includes selective media for the isolation of bacteria, yeast, and molds. Sabouraud agar with the addition of dextrose or maltose (SDA) and antibacterial drugs that prevent bacterial growth and cycloheximide (Actidion^®^) to suppress mold growth are used as selective media for isolating yeast in vitro. The optimal cultivation time of yeast is 48–72 h, at 37 °C in an aerobic atmosphere. Considering that *Candida* spp. is the dominant cause of otomycosis, the use of *Candida*-chromogenic media has been recommended, allowing the differentiation of species of the genus *Candida* based on the discoloration of the colonies, even in the primary isolate. Chromogenic selective media allow the differentiation of three *Candida* species during primo-isolation, where *C. albicans* forms green colonies, *C. tropicalis* forms blue, and *C. krusei* forms pink colonies with a rough appearance [4]. The importance of these media is heightened by the fact that they allow for diagnosing mixed infections caused by different *Candida* species, which is impossible on other media. During mycological analyses, it is important to provide optimal conditions for molds as well. This includes media that contain a higher concentration of carbohydrates and antibacterial drugs, but without action. Moreover, the optimal cultivation time for molds is 7–14 days, at 26–28 °C in an aerobic atmosphere (some mold species grow only at this temperature) [40,41]. If the protocols for culturing fungi, especially molds, are not followed, a certain number of patients may receive a misdiagnosis from the microbiological examination.

Differentiation of yeasts is possible based on tests used to determine the distinctive phenotypic characteristics of different species, using assimilation or other biochemical tests, even automated systems (VITEK^®^-automated system) in more advanced laboratories. Numerous commercial tests for differentiating yeast species and genera, as well as in vitro testing the antifungal susceptibility, are available to many laboratories [Fungifast (ELITech Microbiology Reagents, Puteaux, France) and Fungitest^TM^ (BioRad, Marnes-la-Coquette, France), or Integral System YEASTS Plus (Liofilchem^®^, Roseto degli Abruzzi, Italy)] [4].

Non-dermatophyte molds are identified by their macroscopic and microscopic characteristics during their saprophytic growth phase on nutrient media. However, this identification method requires expert knowledge, which is a significant limitation. 

To overcome the limitations and disadvantages of conventional methods, scientists and researchers are working on developing non-culture methods that will reduce the time between sampling to results. One promising method is MALDI-TOF-MS (matrix-assisted laser desorption and ionisation–time of flight–mass spectrometry, which can currently identify approximately 800 yeast species and 3000 mold species, based on saved reference spectra) [42]. In addition, molecular techniques have also been explored. One study analyzed black *Aspergilli* isolates from otomycosis patients using molecular analyses, and identified a high diversity of species within the *Aspergillus niger* complex. By determining molecular targets for identification, this approach may enable the development of rapid molecular testing for otomycosis [5]. Another study analyzed partial β-tubulin gene sequences of *Aspergillus* molds, and found that *A. niger* and *A. tubingensis* were less susceptible to antifungal drugs compared to other species. Prompt and accurate identification of the infective agents can facilitate the timely selection of the most effective antimycotics [43]. 

Machine learning specifically trained neural network models also promise prompt and accurate diagnosis of otomycosis caused by *Aspergillus* spp. and *Candida* spp. [44]. Machine learning has been applied to medical imaging, for example, through radiological and microscopic images of inflammation cells [45], as well as histological images. It has also been used to recognize and differentiate bacteria and helminthic eggs. Previous studies have shown that macroscopic or microscopic classification of one fungal genus, such as *Aspergillus* spp. [45] and *Leuchorrea* spp. [46], yielded good results using convolutional neural networks (CNNs), but only when the genus was already determined. Recently, a study aimed to develop software for automatic, fast, and efficient diagnosis of otomycosis using an otoendoscopic image gallery to train a neural network model [44]. The authors reported an average accuracy of 92.42%, and suggested that this system could be used in the diagnosis of otomycosis.

## 6. Treatment

The lack of official therapeutic guidelines and protocols has led to disagreement among experts on the optimal treatment for fungal infections. Moreover, the treatment duration is also undefined. Thus, careful consideration of the evidence, patient factors, and clinical judgment are crucial in determining the most appropriate approach [40,47]. Many authors believe it is necessary to identify the causative agent and choose the most effective antimycotic based on the previously determined specific sensitivity. However, other authors suggest that it is advisable to select an appropriate treatment based on the general effectiveness and characteristics of the drug, regardless of the pathogen type [35,48].

In France, nystatin is recommended as the first-line local treatment of otomycosis, typically used in combination with oxytetracycline, polymyxin B, and dexamethasone for up to 15 days [47]. Although this polyene has a wide spectrum of activity against both yeasts and molds, there are inconsistent opinions in the literature about its effectiveness against *Aspergillus* spp., one of the dominant causative agents in otomycosis [35,40]. On the other hand, in the USA, clotrimazole, as topical imidazole, is considered the drug of choice for the treatment of uncomplicated otomycosis [27]. Accordingly, recent research conducted in India suggests that 1% clotrimazole cream applied topically may effectively treat otomycosis [49]. However, some studies with smaller sample sizes do not classify clotrimazole as a first-choice drug due to its lesser effectiveness against *Aspergillus*-otomycosis [32,40,50].

Other antifungal agents, such as miconazole, bifonazole, isoconazole [51] and ciclopiroxolamine, can also be used for the treatment of otomycosis. However, in vitro studies and treatment monitoring have shown that these antifungals have an unequal effect on yeasts and molds, with miconazole being less effective against various species of the genus *Candida* [4,47]. Some studies have also reported good in vitro effectiveness of efinaconazole, lanoconazole, and luliconazole against *Aspergillus* species, but lower susceptibilities of ravuconazole were observed against *A. tubingensis* and *A. niger*, which are the most prevalent causative agents of otomycosis [43]. In a study conducted last year comparing the clinical efficacy of sertaconazole with other imidazoles, satisfactory effects were reported in patients with otomycoses, yet there was no significant difference observed compared to treatment with miconazole and clotrimazole [31]. 

A recent survey conducted in Russia involving over 300 patients with fungal ear infection recommended mandatory mycological control in addition to clinical examination to assess the effectiveness of therapy [52]. The results of this study revealed that terbinafine, naftifine, and chlornitrophenol were the most effective in local treatment for mold-induced otomycosis, while clotrimazole or allylamines (terbinafine and naftifine) were found to be the best choice for *Candida* otomycosis. It was noted that local antifungal treatments should be applied for at least 3–4 weeks with constant laboratory monitoring. Similar findings were reported in a study conducted in Serbia with an analogous design, which showed that nystatin and naftifine yielded better results in the treatment of patients with *Aspergillus*-otomycosis compared to clotrimazole [40]. 

These results support the potential effectiveness of allylamines, such as terbinafine and naftifine, in the treatment of otomycosis caused by yeasts or molds, as was also suggested in a recent study by Ting-Hua Yang, who demonstrated the non-toxicity of terbinafine to the inner ear end organs at a dosage of 0.4 mg [53].

Although various antiseptics, acidifying agents, and medications containing anti-infective agents with corticosteroids have been investigated for the treatment of otomycosis, there is insufficient evidence to support their increased efficacy compared to antimycotic drugs, as reported by Khrystyna Herasym in 2016 [54]. Moreover, extensive ear canal debridement followed by topical antifungal medications is recommended for patients with a noninvasive form of infection.

Another important fact is that antifungal ointments have distinct advantages over liquid formulations. Creams, due to their higher viscosity, remain longer on the skin’s surface and are considered safer for use in patients with perforated eardrums, as there is a lower chance of the drug entering the middle ear [55]. One recommendation for liquid formulations of antimycotics is to soak cotton wool or gauze pads in the solution and leave them in the ear canal for 5–10 min, 2–4 times a day [52]. In addition to all the described treatment modalities, it is crucial to restore the physiological conditions in the ear canal by avoiding excessive use of topical medications and protecting the ear canal from further damage, which can disrupt local homeostasis [25].

Systemic antimycotics, such as triazole antifungal drugs (fluconazole, itraconazole, voriconazole, posaconazole), may be used in severe forms of otomycosis, or when previous local therapy has been ineffective [48]. These drugs are effective against infections caused by fungi of the genera *Candida* and *Aspergillus,* and are essential in the treatment of complex forms of the disease, especially when there are complications such as mastoiditis and meningitis [56].

However, an investigation of in vitro antifungal susceptibility demonstrated that non-*albicans Candida* species, as causative agents of otomycosis, might have varying sensitivity in a dose-dependent manner to fluconazole (MIC = 32 mg/mL). Moreover, it was proven that *C. krusei*, besides fluconazole, also had lower sensitivity to itraconazole (MIC = 0.5 mg/mL) [4]. This highlights the importance of identifying specific causative agents and conducting susceptibility testing to guide the appropriate choice of systemic antifungal therapy in severe cases of otomycosis.

## 7. Epidemiology

From an epidemiological perspective, it can be stated that the climatic conditions of tropical and subtropical regions, characterized by long summers and rainy seasons, are the most common risk factors for the occurrence of otomycosis. A wide spectrum of different species and genera of fungi causing otomycosis has been isolated and identified in these areas. Specifically, numerous yeast and mold species have been identified in Asia [2,57,58], Africa [10], Central America [12], and South America [59] (Table 1, Figure 1). Surveys conducted in Europe have not noted any seasonal influence [4]. 

The age and gender distribution of patients with otomycosis exhibits inconsistent findings across various reports. Some studies have stated that men are more frequently affected than women [4,24], but this is not uniform in all surveys [1,14]. Similarly, the fact that people between the ages of 25 and 45 are most often affected [1,10,12,24] is accompanied by findings that the prevalence of the disease increases with ageing [4].

Risk factors for EAC fungal infection include factors that damage the mechanical barrier of the EAC skin. In this group, frequent use of headphones [40], using hard objects to scratch the ear [11,15], and injuries to the skin [24] are dominant factors. Besides damage to the skin as a mechanical barrier, factors that disturb the EAC skin microbiota and affect the reduction of cerumen [21,24] can also impair non-specific resistance. Overuse of chemicals such as soaps, shampoos, boric acid, povidone–iodine, hydrogen peroxide, and other antiseptics, as well as excessive use of antibacterial ear drops, can disrupt the balance of the microbiota in the EAC. This disruption often leads to a reduction in the number of bacteria, which in turn can result in the overgrowth of fungi and increased risk of fungal infections [2,15,40]. Additionally, the use of coconut oil [11] and mustard oil [2] as an alternate treatment or for alleviating symptoms and signs of infection has been identified as a possible risk factor for otomycosis. Furthermore, clothing habits such as wearing head coverings, e.g., a turban or veil [6,61], have been highlighted as risk factors in areas where traditional clothing is common or recommended. Frequent swimming, recreationally or professionally, as well as the use of saunas and other spa treatments, can also predispose a person to this infection.

Lastly, preexisting diseases and conditions such as diabetes [1,14], immunosuppressive therapy [63], prolonged systemic antibiotic treatment [63], a history of otologic procedures [14], retroviral infection [1], sinonasal and nasopharyngeal malignancy [1], and mastoid cavity after surgical procedures [14] may predispose a person to the development of this infection. In premature newborns, intubation, artificial ventilation, and respiratory distress have also been identified as significant risk factors for fungal infection of the EAC [63].

## 8. Conclusions

Similarly to other superficial fungal infections, otomycosis is generally a benign condition that does not threaten life. However, frequent cases of chronicity, the high prevalence of these infections, reductions in the quality of life of affected individuals, and rare complications (such as perforation of the tympanic membrane or disseminated infection involving bones, nerves, and even the central nervous system) indicate the need for a new strategy for diagnosis and treatment.

Based on our knowledge and the results of the conducted studies, the following highlights emerge. (i) In cases of chronic external auditory canal inflammation, laboratory analyses are recommended. (ii) Microbiological examination, in addition to bacteriological analysis, should include mycological analysis, which encompasses procedures for isolating both yeasts and molds. (iii) For accurate interpretation of the results, serial cultures (up to three) may be necessary to differentiate between pathogenic and saprophytic fungi. (iv) Treatment selection, including the choice of antimycotics and method of application, should be based on specific fungal causative agents. However, systemic antifungal drugs should only be considered for cases involving immunodeficiency. Official protocols and recommendations regarding diagnosis and treatment can significantly assist in selecting the most effective medication and treatment duration. 

Avoidance or reduction is recommended for certain risk factors, including activities such as frequent recreational or professional swimming, the use of saunas and other spa treatments, alternate treatment application, and clothing habits, as they have the potential to damage the mechanical barrier or microbiota of the skin in the EAC, leading to a reduction in cerumen production. Meanwhile, preexisting diseases and conditions such as diabetes, immunosuppressive therapy, prolonged systemic antibiotic treatment, and a history of otologic procedures can serve as potential predictors for the development of EAC fungal infections.

## Figures and Tables

**Figure 1 jof-09-00662-f001:**
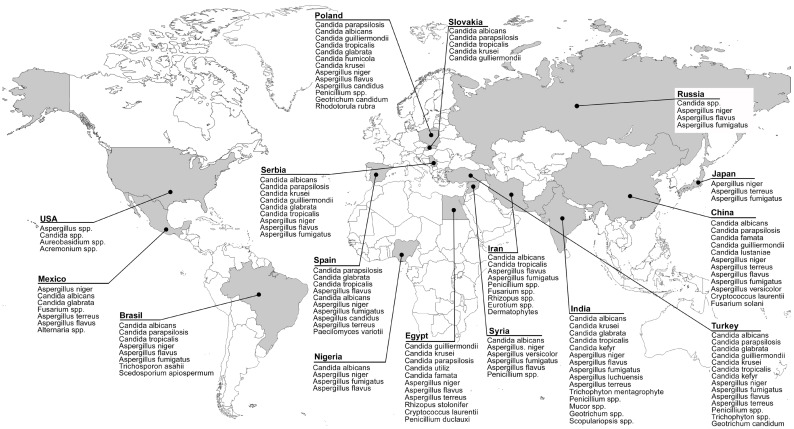
Distribution of otomycosis causative agents.

**Table 1 jof-09-00662-t001:** Species distribution, clinical manifestations, and risk factors for otomycosis.

Country (Sample Size *)	Identified Fungal Species **	Reported Signs and Symptoms ***	Predisposing Factors ***	References
**Asia**				
**Japan** (29 patients)	*A. niger*, *A. terreus*, *A. fumigatus*	N/A	N/A	Hagiwara, S. et al. [43]
**China** (108 patients)	*A. niger*, *C. albicans*, *A. terreus*, *A. flavus*, *C. lustaniae*, *A. fumigatus*, *C. parapsilosis*, *C. famata*, *Cryptococcus laurentii*, *C. guilliermondii*, *A. versicolor*, *Fusarium solani*	Pruritus, Otorrhea, Ear fullness, Hearing loss, Otalgia, Tinnitus	N/A	Jia, X. et al. [58]
**India** (30 patients)	*A. niger*, *A. flavus*, *C. tropicalis*	Pruritus, Otalgia, Ear fullness, Hearing loss, Otorrhea	Ear pricking with hard objects, Use of oil ear drops, Swimming or pond baths, Diabetes, Immunodeficiency	Panigrahi, M. et al. [36]
**India** (350 patients)	*A. niger*, *A. flavus*, *A. fumigatus*, *C. albicans*, *C. krusei*, *C. tropicalis*, *Penicillium* spp., *Mucor* spp., *Trichophyton mentagrophyte*	Hearing loss, Ear fullness, Pruritus, Otalgia, Otorrhea, Tinnitus	Ear pricking with hard objects, Use of oil ear drops	Agarwal, P. et al. [2]
**India** (100 patients)	*A. niger*, *A. flavus*, *A. fumigatus*, *C. albicans*, *C. tropicalis*, *C. glabrata*,*C. kefyr*, *Penicilium* spp., *Geotrichum* spp., *Scopulariopsis* spp.	Pruritus, Otalgia, Ear blockage, Tinnitus, Hearing loss, Otorrhea	Use of oil ear drops, Use of antibiotic or wax-dissolving ear drops, Ear pricking with hard objects, Diabetes, Swimming	Rawat, S. et al. [15]
**India** (100 patients)	*A. niger*, *A. fumigatus*, *A. flavus*, *Penicillium* spp., *C. albicans*, *Rhizospus* spp., *Chrysosporium* spp.	Pruritus, Ear fullness, Otorrhea, Otalgia, Tinnitus	Use of oil ear drops, Ear pricking with hard objects, Use of antibiotic ear drops	Prasad, S.C. et al. [11]
**Pakistan** (180 patients)	N/A	Hearing loss, Pruritus, Otalgia, Otorrhea, Tinnitus	N/A	Anwar, K. et al. [25]
**Iran** (129 patients)	*A. niger*, *A. flavus*, *A. fumigatus*, *C. albicans*, *Penicillium* spp., Dermatophytes, *C. tropicalis*, *Fusarium* spp., *Rhizopus* spp., *Eurotium* spp.	Pruritus, Ear fullness, Otorrhea, Otalgia	Ear pricking with hard objects, Diabetes	Kazemi, A. et al. [57]
**Syria** (70 patients)	*A. niger*, *C. albicans*,*A. versicolor*, *A. fumigatus*, *A. flavus*, *Penicillium* spp.	Otorrhea, Otalgia, Hearing loss	N/A	Ismail, M.T. et al. [60]
**Turkey** (87 patients)	*A. niger*, *A. fumigatus*, *A. flavus*, *A. terreus*, *C. albicans*, *C. tropicalis*, *C. kefyr*	Pruritus, Otalgia, Hearing loss, Tinnitus, Otorrhea	Wearing a traditional head covering, Swimming in the pool/sea, Spa baths, Itching on other body parts, Long-term antibiotic treatment, Rainy weather season	Ozcan, K.M. et al. [61]
**Turkey** (544 patients)	*A. niger*, *C. tropicalis*, *A. fumigatus*, *C. albicans*, *A. terreus*, *A. flavus*, *C. parapsilosis*, *C. glabrata*, *Penicillium* spp., *C. kefyr*, *C. guilliermondii*, *C. krusei*, *Trichophyton* spp., *Geotrichum candidum*	N/A	N/A	Değerli, K. et al. [8]
**Africa**				
**Egypt** (110 patients)	*A. niger*, *A. flavus*	Pruritus, Otalgia, Hearing loss, Otorrhea	Ear canal trauma, Swimming, Use of antibiotic ear drops, Absent cerumen, Summer season	Abdelazeem, M. et al. [24]
**Egypt** (102 patients)	*A. niger*, *A. flavus*, *C. famata*, *A. terreus*, *C. parapsilosis*, *C. utiliz*, *Rhizopus stolonifer*, *C. guilliermondii*, *C. krusei*, *Cryptococcus laurentii*, *Penicillium duclauxi*	Pruritus, Otalgia, Otorrhea, Hearing loss, Tinnitus	N/A	Ali, K. et al. [10]
**Nigeria** (378 patients)	*A. niger*, *A. fumigatus*, *C. albicans*, *A. flavus*	Pruritus, Otalgia, Tinnitus, Ear fullness, Hearing loss, Otorrhea	Ear pricking with hard objects, Long-term use of oral antibiotics, Use of antibiotic ear drops, Diabetes, Sino-nasal and nasopharyngeal malignancy, Retroviral infection	Fasunla, J. et al. [1]
**Central America**				
**Mexico** (40 patients)	*A. niger*, *C. albicans*, *C. glabrata*, *Fusarium* spp., *A. terreus*, *A. flavus*, *Alternaria* spp.	Pruritus, Hearing loss, Otorrhea, Otalgia	N/A	Alarid-Coronel, J. et al. [12]
**North America**				
**USA** (132 patients)	*Aspergillus* spp. *Candida* spp. *Aureobasidium* spp. *Acremonium* spp.	Otalgia, Otorrhea, Hearing loss, Ear fullness, Pruritus, Tinnitus	Diabetes, History of otologic procedures, Mastoid cavity after surgical procedure	Ho, T. et al. [14]
**South America**				
**Brasil** (20 patients)	*C. albicans*, *C. parapsilosis*, *A. niger*, *A. flavus*, *A. fumigatus*, *C. tropicalis*, *Trichosporon asahii*, *Scedosporium apiospermum*	Pruritus, Otalgia, Otorrhea, Hearing loss	Chronic otitis, Long-term antibiotic treatment, Absent cerumen, Ear pricking with hard objects	Pontes, Z.B. et al. [59]
**Europe**				
**Poland** (96 patients)	*C. parapsilosis*, *C. albicans*, *A. niger*, *A. flavus*, *Penicillium* spp., *C. guilliermondii*, *C. tropicalis*, *C. glabrata*, *C. humicola*, *C. krusei*, *A. candidus*, *Geotrichum candidum*, *Rhodotorula rubra*	Pruritus, Otorrhea, Ear fullness, Hearing loss, Tinnitus, Otalgia, EAC swelling and redness, Headache	N/A	Kurnatowski, P. et al. [62]
**Slovakia** (40 patients)	*C. albicans*, *C. parapsilosis*, *C. tropicalis*, *C. krusei*, *C. gulliermondii*	Burning sensation in the ear, Pruritus, Ear fullness, Hearing loss, Otalgia, Otorrhea, Tinnitus, Headache, Nausea	Swimming pool and sauna usage, Diabetes, Immunosuppressive therapy, Long-term antibiotic treatment	Dorko, E. et al. [63]
**Spain** (390 patients)	*C. parapsilosis*, *A. flavus*, *C. albicans*, *A. niger*, *A. fumigatus*, *A. candidus*, *A. terreus*, *C. glabrata*, *C. tropicalis*, *Paecilomyces variotii*	Pruritus, Otalgia, Hearing loss	N/A	García-Agudo, L. et al. [13]
**Russia** (331 patients)	*A. niger*, *A. flavus*, *A. fumigatus*, *Candida* spp.	N/A	N/A	Kryukov, A.I. et al. [52]
**Serbia** (292 patients)	*A. niger*, *C. albicans*, *C. parapsilosis*, *A. flavus*, *A. fumigatus*, *C. krusei*, *C. guilliermondii*, *C. glabrata*, *C. tropicalis*	N/A	N/A	Tasić-Otašević, S. et al. [4]
**Serbia** (30 patients)	*A. niger*, *A. flavus*	N/A	Use of antibiotic ear drops, Use of EAC hygiene pro-ducts, Headphone usage, Frequent ORL examinations and rinsing of EAC, Corticosteroid ear drop usage, Predisposing diseases	Bojanović, M. et al. [40]

* Number of patients diagnosed with otomycosis, ** Sorted by number of isolates, *** Sorted by frequency of occurrence, N/A Not Available.

## Data Availability

No new data were created or analyzed in this study. Data sharing is not applicable to this article.

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
