# Peer review of "Etiology, Predisposing Factors, Clinical Features and Diagnostic Procedure of Otomycosis: A Literature Review"

_jof, 2023, doi:10.3390/jof9060662_

Round 1

Reviewer 1 Report

This is a comprehensive review and the authors are to be applauded for the large amount of time and study they put in to this paper.

To call a fungus a pathogen, rather than a saprophyte, serial cultures (up to 3)  may need to be done especially when Aspergillus and Candida species are found. In the skin by biopsy or KOH, pseudohyphae rather than yeast forms ,are more associated with pathogenicity .

Candida species and Aspergillus are commonly found on skin cultures and usually are saprophytes.

The authors correctly point out that Sabs agar with chloramphenicol and cyclohexamide  and Sabs agar with just chloramphenicol should be utilized on each fungal culture.

In general on the skin, allylamines work less well on Candida than the azoles.

Antifungals may have a weak anti inflammatory effect equivalent to a 1% hydrocortisone.

For dermatologists, seborrheic dermatitis /psoriasis likely plays a greater role than fungi.

Scratching breaks the skin barrier, as do seborrheic  dermatitis/psoriasis , and allows for secondary infection usually bacterial, but sometimes fungus.

The disorder commonly itches, but pain, pus,oozing , suggests a secondary infection, more often bacterial than fungal.

So , even though there are not enough papers in the literature suggesting a primarily inflammatory etiology of this disorder, from a skin/ dermatology point of view, there should be. 

It is more likely than not, many cases of diagnosed “otomycosis”, is in fact primarily inflammatory and not infectious. Seborrheic dermatitis, psoriasis, and contact dermatitis (to neomycin, etc.) Are the inflammatory disorders of the ear canal most commonly seen, even though the dermatology literature does not adequately address this subject.

Author Response

Dear Reviewer,

We sincerely appreciate your recognition of our paper as a comprehensive review, as well as acknowledging the substantial time and effort dedicated to its development. Furthermore, we would like to express our gratitude for your suggestions and guidance, particularly concerning crucial aspects related to the diagnosis and treatment of otomycosis.

Your comments were very useful and we have incorporated all the omitted important facts pertaining to the assessment of pathogenic potential of fungi and interpretation of mycological analyses.

It is indeed a significant challenge in medical mycology to differentiate fungal causative agents from fungal microbiota or transient fungal flora through mycological analyses. As recommended, we have included serial cultures (up to 3) to aid in the interpretation. This approach has proven effective in addressing this issue in various forms of fungal infections.

Regarding the potential of certain fungi to cause infections and their pathogenicity, a similar situation arises in the case of otomycosis. To date, an unexplained aspect persists: whether the fungi themselves initiate inflammation and infection, or if they predispose skin diseases that disrupt the mechanical barrier and create conditions for the development of bacterial or fungal infections. We have included this consideration in our paper.

All changes are marked in yellow.

Sincerely Prof. Otašević

Reviewer 2 Report

This manuscript reviewed the important features of otomycosiswith its pathogenesis, clinical presentation, diagnosis and treatment, and epidemiological features. In view of the high prevalence of otomycosis worldwide and the lack of an officially recommended diagnostic process for its treatment, the review will help for providing guidance on the diagnosis and management of otomycosis.

There are some contents need to be added.

1. In Materials and methods, it describes the results taken by the authors to search the literature and the work of the different authors involved in writing this manuscript, but does not apply the database to make a new analysis of the epidemiological survey of otomycosis. Are there better statistical and comparative methods to complement the results?

2. The advantages, disadvantages and applicability of the method s mentioned in the manuscript are not well summarized. It is suggested to add relevant contents.

3.     It should be prospected the development of clinical diagnosis and treatment of otomycosis, based on the summary of epidemic characteristics, susceptibility factors and other information.

Author Response

Dear Reviewer,

We would like to express our gratitude for acknowledging that our manuscript thoroughly reviewed the significant aspects of otomycosis. We appreciate your recognition that the review will serve as a valuable resource in guiding the diagnosis and management of otomycosis. Your suggestions have been incorporated, resulting in significant improvements to our paper.

  1. We have included additional explanations in the Materials and Methods section regarding our methodology. Specifically, we have enhanced the description of how the data was compared in a narrative manner, as statistical pooling of the studies was not feasible due to heterogeneity in study designs, patient characteristics, diagnostic procedures, treatment modalities, and outcomes.
  2. In the Materials and Methods section, we provided a concise overview of the advantages and disadvantages of employing a narrative review as our method of analysis. Notably, this approach allows for a comprehensive and precise summary of the existing knowledge on the subject, providing a solid foundation for future research. However, when assessing a narrative review, it is crucial to remain aware of the potential for selection bias.
  3. Based on our current knowledge and the findings from conducted studies, we have emphasized and provided certain guidelines for the prospective management of otomycosis, including clinical evaluation, laboratory diagnosis, and treatment approaches. Additionally, we have summarized the epidemic characteristics and identified susceptibility factors in the concluding section.

All changes were marked blue.

Sincerely Prof. Otašević